# Lifestyle patterns and their nutritional, socio-demographic and psychological determinants in a community-based study: A mixed approach of latent class and factor analyses

Mahdi Vajdi[1], Leila Nikniaz[2], Asghar Mohammad Pour Asl[3], Mahdieh Abbasalizad Farhangi[4]*

1 Student Research Committee, Tabriz University of Medical Sciences, Tabriz, Iran, 2 Tabriz Health Services Management Research Center, Health Management and Safety Promotion Research Institute, Tabriz University of Medical Sciences, Tabriz, Iran, 3 Department of Epidemiology and Biostatistics, Health and Environment Research Center, Faculty of Health, Tabriz University of Medical Sciences, Tabriz, Iran, 4 Drug Applied Research Center, Tabriz University of Medical Sciences, Tabriz, Iran

* abbasalizad_m@yahoo.com

**Data Availability Statement:** All relevant data are within the paper and its Supporting Information file.

## Abstract

### Background

Lifestyle risk factors, such as unhealthy diet, physical inactivity or tobacco smoking can have detrimental effects on health and well-being. Therefore, it is important to examine multiple lifestyle risk factors instead of single ones. Cluster analysis allows the combination of single health behaviors in order to recognize distinguished behavior patterns. This study aimed to evaluate lifestyle patterns of general adult population in northwest of Iran with particular focus on dietary patterns, physical activity, and smoking status.

### Methods

The current cross-sectional study consists of 525 adults aged 18–64 years from East-Azarbaijan Iran. Latent class analysis (LCA) was applied to recognize patterns of lifestyle behaviors with ingredients of diet, physical activity, and smoking status. Dietary intake was assessed using a validated food frequency questionnaire and dietary patterns were derived using factor analysis. Biochemical parameters including fasting blood sugar (FBS), serum lipids, liver enzyme and serum 25(OH)-D3 were measured with commercial ELIZA kits.

### Results

Mean ages of participants were 42.90 ± 11.89 years. Using principal component analysis (PCA) three major dietary patterns were extracted including traditional dietary pattern (e.g. nuts and dry fruits), unhealthy dietary pattern (e.g. fast foods, refined grains) and the healthy dietary patterns (e.g. fruits, vegetables). Using LCA, three classes of lifestyles pattern were identified: 1st class was characterized by a healthy dietary pattern, moderate physical activity, and low probability of smoking. 2nd class was characterized by a traditional dietary pattern, low level of physical activity and low probability of smoking and 3rd class was

**Funding:** The work has been granted by Research Undersecretary of Tabriz University of Medical Sciences (Registration number: IR.TBZMED. REC.1398.07) and is obtained from the M.S thesis of Mahdi Vajdi. The funders had no role in study design, data collection and analysis, decision to publish, or preparation of the manuscript.

**Competing interests:** The authors have declared that no competing interests exist.

**Abbreviations:** BMI, Body mass index; PA, Physical activity; PCA, Principal Components Analysis; IPAQ, International physical activity questionnaire; FFQ, Food frequency questionnaire; MET, Metabolic equivalent; BIC, Bayesian information criterion; AIC, Akaike information criterion; SD, Standard Deviations; WHR, waist hip ratio.

characterized by a unhealthy dietary pattern, low level of physical activity and low probability of smoking and further analysis found that there were significant differences in body mass index (BMI), Waist-to-hip ratio (WHR), FBS, Hemoglobin (Hb), education levels and anxiety status between classes (P <0.05).

## Conclusion

This study attempts to classify Iranian adults by their own health behavior. Healthcare professionals should be aware of associations between different lifestyle risk factors and health promotion strategies should further focus on multiple behaviors at the same time. In our country, more studies about the adult population are needed to support the observed findings of our study and therefore allow for a certain generalization of the observations.

## 1. Introduction

Modifiable lifestyle risk factors, such as physical inactivity, unhealthy diet, as well as tobacco smoking are associated with mortality and development of numerous chronic diseases, such as type-2 diabetes, cardiovascular diseases, hypertension and various types of cancer [1, 2]. It has been estimated that 53% of deaths in adults are due to unhealthy lifestyles behaviors such as extreme alcohol intake, smoking, physical inactivity and poor diet; as a consequent, lifestyle risk factors cause considerable costs for society and health care systems [3, 4]. Numerous parameters including socio-demographic determinants influence the ingredients of life style patterns; for example, unhealthy behaviors such as sleep deficiency, physical inactivity, unhealthy diet, and tobacco smoking are more prevalent amongst younger age groups, unmarried men, and those who have a poor level of education [5–7]. An unhealthy diet is an important risk factor for almost all of the chronic disease and several studies have highlighted the role of dietary patterns rather than a single nutrient as an important determinant of chronic diseases [8, 9]. Lower intake of fruits and vegetables, insufficient physical activity, high consumption of salt and sugars and smoking are highly related to micro- and macro vascular problems [10, 11]. Increased physical activities, weight management, smoking cessation are important preventive strategies against cardiovascular events [12]. Most of the studies have focused on the each of the ingredients of life style separately while; underlying patterns and combinations of lifestyle behaviors in Iranian adults are not studied well. Lifestyle risk factors including unhealthy diet, physical inactivity, smoking and excessive alcohol consumption are correlated with each other and these unhealthy behaviors are not randomly distributed in the population and occur in combination with other behaviors. Recently, investigators have attempted to cluster lifestyle risk factors. Combinations of two or more unhealthy lifestyle behaviors are usually related to a higher increased risk of disorders than can be expected from each of the risk factors alone [13, 14]. The lifestyle clusters in a community can be related to different patterns of social and demographic risk factors [15, 16]. Therefore, it is particularly important to examine the combination of lifestyle risk factors to identify the different lifestyle patterns and possible synergistic effects of these risk factors rather than the effect of only one single behavior.

Latent-class analysis (LCA) is a person-centered statistical technique that helps to analysis population heterogeneity within observed characteristics through the identification of underlying subgroups of individuals [17]. It is an appropriate method for dichotomous and discrete

variables and focus on the association between individuals within the sample [17]. One important advantage of LCA is that multiple features that categorize persons into subgroups can be surveyed concurrently instead of examining every one of these characteristic separately. Some studies, report that LCA can be a more useful method to describing health behaviors than a summary score [18–20]. Using LCA method, Schnuerer et al [21] identified three classes of health conscious, substance use and non-exercising overweight among job-seekers. In the current study we aimed to evaluate the integrative role of life style determinants including dietary patterns, physical activity and smoking on demographic and biochemical parameters among apparently healthy population of East Azerbaijan -Iran.

## 2. Methods and materials

### 2.1. Sample and procedure

This study was conducted within the framework of the major lifestyle promotion project (LPP). LPP was a cross-sectional population based study conducted in urban and regional parts of East Azerbaijan -Iran. The sampling was conducted with the probability proportional to size multistage stratified cluster sampling [22]. In probability proportional to size sampling, the selection probability for each element is set to be proportional to its size measure. Briefly, the postcode was used as the sampling frame and the clusters were selected according to postcode in the current study [23]. Every address in this system was summarized in a 10-digit postcode number. Clusters comprise one to numerous blocks or parts of blocks. Blocks were usually attached buildings. After determining the cluster start point, enrollment and data collection was started. 150 clusters were selected. In each cluster, 4 participants (18–64 years) were enrolled (600 participants). This started from the household at the cluster start point and continued toward the other households until the required number of participants enrolled. Finally after excluding incomplete questionnaire 525 samples were included in statistical analysis. Subjects were included in the current study if their original nationality was Iranian and aged between 18 and 64 years. Subjects with physical disability, severe chronic illness requiring bed rest, mental disability, active liver injury, and pregnant women were not included in the study. Informed written consent was obtained from all subjects.

### 2.2. Demographic and anthropometric variables

The socio-demographic information included age, educational level, employment status, gender, family size, marital status and residential area were collected through questionnaire. Blood pressure was measured with a standard manual mercury sphygmomanometer in sitting position. Weight was measured while participants were minimally dressed without shoes by calibrated Seca scale (Dubai, United Arab Emirates) with the precision of 0.1 kg. Height was measured without shoes using a stadiometer with a precision of 0.1 cm in a standing position. The BMI was calculated as weight (kg) divided by height $^2$. Waist circumference (WC) was measured at the minimum circumference between the iliac crest and the rib margin without any pressure to the body, with the precision of 0.1 cm. WHR was calculated as WC was divided by hip circumference. Anxiety was evaluated using the Generalized Anxiety Disorder 7-item scale (GAD-7). The GAD-7 is seven-item standardized tools that evaluate the severity of general anxiety with increasing scores shows increasing severity of anxiety [24]. This scale evaluates the frequency of general anxiety disorder symptoms over the previous two weeks on a 4-point Likert scale (not at all = 0, several days = 1, more than half the days = 2, nearly every day = 3). Scores of 0–5, 6–9, 10–14 and ≥15 specify normal, low, moderate and high [25].

## 2.3. Biochemical tests

After a 12–14 h overnight fasting, 10 ml venous blood samples were collected from all participants and placed in vacutainer/ siliconized test tubes without anticoagulant for biochemical determinations. The samples were immediately centrifuged at 2000 rpm for 10 min at room temperature and the serum samples were separated. Serum triglyceride (TG) (Pars Azmoon kit No. 03 1500), total cholesterol (Pars Azmoon kit No. 010 1500), high-density lipoprotein cholesterol (HDL-C) (Pars Azmoon kit No. 012 1500) and fasting blood glucose (FBG) (Pars Azmoon kit No. 017 1500) were measured by enzymatic colorimetric methods (Pars Azmoon kit, Tehran, Iran) using an automatic analyzer (Abbott, model Alcyon 300, USA). Aspartate aminotransferase (AST) (Pars Azmoon kit No.018 1500) and Alanine aminotransferase (ALT) (Pars Azmoon kit No.019 500) were measured using the ultraviolet method [26, 27]. LDL-C (Pars Azmoon kit No. 022 1050) was calculated by Friedewald equation [28]. Chemiluminescent immunoassay technology was used for the quantitative determination of serum 25 (OH) D (25 OH Vitamin D Total Assay, DiaSorin, Saluggio, Italy, catalog no. 26100).

## 2.4. Measurement of the variables included in the latent class analysis

Lifestyle patterns, in the current study, are the underlying latent variables including physical activity, anxiety status, dietary patterns and smoking. The indicators were chosen to ensure conditional independence of variables within each latent class [29]. Binary indicators (healthy versus unhealthy behavior) were created based on existing recommendations and/or propositions by the literature, as described below. Binary indicator for smoking was created (never smoked versus current and former smokers) [30–32].

**2.4.1. Physical activity (PA).** Information on physical activity was collected using the Persian short form of International Physical Activity Questionnaire (IPAQ-SF), through a face-to-face interview and the validity of the translated form of this questionnaire was tested in the prior study on Iranian adult population [33]. IPAQ-SF consists of seven questions asking about how much time the respondent has spent doing each activity, during the last week; and based on these data, the metabolic equivalents (MET -minutes/week) is calculated with the MET minute scores being equivalent to kilocalories expended per week. The volume of activity was calculated by weighting each type of activity according to its energy requirements to yield a score in MET-minutes [34]. Computation of the total score of physical activity, expressed in metabolic equivalents (MET-minutes/week), requires the summation of duration (in minutes) and frequency (days) of walking (low intensity), moderate-intensity and high-intensity activities. Regarding the scoring system provided by the IPAQ, participants were categorized as having high, moderate, or low level of activity. High PA was defined as 8.0 METs or more, moderate PA was identified as 4.0 METs, and low was defined as 3.3 METs [34].

**2.4.2. Dietary patterns.** Usual dietary intake was evaluated using a validated 80-item quantitative food-frequency questionnaire (FFQ) [35]. The questionnaire was completed as face-to-face personal interviews by trained dietitians specifying the frequency of intake in a daily, weekly, monthly or yearly basis. The reported frequency of each item was then converted to a daily consumption. The FFQ food items then were categorized into 23 food groups according to their nutrient content to include in the factor analysis (Table 1).

**2.4.3. Smoking status.** Smoking status data was obtained by Global Tobacco Surveillance System (GTSS) consisting thirty items with a face-to-face interview [36]. Tobacco consumption was defined as daily smoking of at least one cigarette, cigar, pipe or cigarillo within the last 30 days and a binary indicator for smoking was created (never smoked versus current and former smokers).

Table 1. Components food groups used in dietary pattern analysis.

| Food group | Food item |
|---|---|
| Fruits | Apples, banana, raisin, cantaloupe, watermelon, melon, sloe, apricot, cherries, sour, fig, plum, nectarine, apricots, pear, peach, lemons, tangerine, oranges, mulberry, kiwi, grape, pomegranate, date, strawberry, grape fruit, persimmon, other fruits |
| Dry fruits | Dried figs, dried dates and all other dried fruits |
| Fruit juices | apple juice, orange juice, other fruit juices |
| Vegetables | Spinach, squash, lettuce, green leafy vegetables, cucumber, cauliflower, green pepper, mixed vegetables, green beans, eggplant, celery, cabbage, onions, garlic, kale, carrots, turnip, corn, tomatoes, green peas, mushroom and all other vegetables |
| Whole grains | Dark Iranian bread (lavash, taftun, sangak and barbari), popcorn, bulgur, barley bread and wheat germ |
| Legumes | Bean, broad beans, peas, lima beans, chickpea, lentil, split pea, soy and other cereals |
| Refined grains | White breads (baguettes, lavash), pasta, milled barley, white flour, rice, starch, toasted bread, biscuits, noodles |
| Sweet and desserts | Cookies, sugar, jam, cakes, pastry, chocolates, honey, confectioneries |
| Red meat | Lamb, beef |
| Organ meat | Kidney, liver, and heart |
| Poultry | Chicken |
| Fish | Canned tuna and any types of fish |
| Fast foods | Sandwich, pizza, French fries, potato chips |
| Egg | Eggs |
| Nuts | Roasted seeds, walnuts, pistachios, peanuts, almonds, hazelnuts |
| Dairy products | Cheese, yoghurt and all types of milk |
| Animal fats | Hydrogenated animal fat |
| Liquid vegetable oil | Canola oil, sunflower oil, and all other vegetable oils |
| Tea | Tea |
| Soft drinks | Soft drinks |
| Coffee | Coffee |
| Salt | Salt |
| Pickle | Pickle |

## 3. Statistical analyses

### 3.1. Descriptive statistics

All statistical analyses was done by using IBM Statistical Package Software for Social Science (SPSS), version 21 (SPSS Inc., Chicago, IL, USA). Baseline characteristics of participants were expressed as means ± standard deviation for quantitative variables, and frequency and percentages for categorical variables. The differences in continuous variables across different class of lifestyle pattern were determined using one-way ANOVA. The χ2 test was used to compare the categorical variables across classes.

### 3.2. Factor analysis

The principal component analysis (PCA) was used based on 23 food groups to extract major dietary patterns (Table 1). The PCA is a method performed to decrease a set of features, through a base change that reduces the correlation between modified features [37]. The factors were rotated by orthogonal transformation (varimax rotation) to retain uncorrelated factors and enhance interpretability. Factor analysis with varimax rotation is used to develop a more

**Table 2. Factor-loading matrix for extracted major dietary patterns.**

| Food groups | Healthy pattern | Unhealthy pattern | Traditional pattern |
|---|---|---|---|
| Fruits | 0.52 | _ | _ |
| Dry fruits | _ | -0.35 | 0.37 |
| Fruit juices | 0.56 | _ | _ |
| Vegetables | 0.42 | _ | _ |
| Whole grains | _ | 0.46- | _ |
| Legumes | 0.43 | _ | _ |
| Refined grains | _ | 0.38 | _ |
| Sweet and desserts | _ | 0.47 | _ |
| Red meat | _ | _ | 0.32 |
| Organ meat | _ | _ | 0.55 |
| Poultry | _ | _ | 0.48 |
| Fish | _ | _ | 0.36 |
| Fast foods | _ | 0.47 | _ |
| Egg | _ | 0.39 | _ |
| Nuts | _ | _ | 0.56 |
| Dairy products | 0.55 | _ | _ |
| Animal fats | _ | 0.46 | _ |
| Liquid vegetable oil | _ | -0.41 | _ |
| Tea | - 0.23 | _ | _ |
| Soft drinks | _ | 0.30 | 0.46 |
| Coffee | - 0.26 | _ | _ |
| Salt | _ | 0.27 | _ |
| Pickle | _ | _ | 0.26 |

interpretable solution. The varimax rotation was used because it has the potential to minimize the number of components. To determine the number of factors, we considered eigenvalues (> 1), and factor interpretability [38]. Considering data reduction in factor analysis, 3 factors were derived. By interpretation of the data and considering the previous literature [39–41], the derived dietary patterns were labeled. The factor score for each pattern was calculated by summing consumption of all food groups weighted by their factor loadings and for each dietary pattern, participants received a factor score. Table 2 shows factor loading of 3 major dietary patterns extracted by factor analysis.

### 3.3. Latent class analysis

The LCA analyses were conducted by using a proc. LCA in SAS 9.2 software. For performing an LCA, two dichotomous and two three-level observable indicators: dietary patterns (3 response categories), physical activity (3 response categories) and smoking status (2 response categories) were used for evaluating lifestyle pattern as a latent variable [42]. The LCA is used to identify the latent classes and unknown patterns based on a set of observed variables from multivariate classified data [43, 44]. Like factor analysis, the LCA can be used to categorize the individuals according to the maximum probability of being in a class. The outcomes of LCA include the number of latent classes, the probability of each indicator in each class. Therefore, membership in each subgroup is based on the similarities in responses to questions related to a set of observed behaviors [45]. To select a suitable number of groups and maximize model fit, first a two-class model was examined. Following, classes were added to the model till no more theoretical and statistical enhancements were observed. We tested models with up to six

classes. Due to several alterations for the number of the latent variable's identified classes, and by comparing the observed response patterns' frequencies with the expected response patterns, the LCA determines the best model and calculates G2. G2 statistic, Bayesian Information Criterion (BIC) and Akaike Information Criterion (AIC) were used to select the best model. For all information criteria, a smaller value represents a more optimal balance of model fit and parsimony; therefore, the minimum BIC or AIC value indicates the best fit model [46, 47] and the class with participants showing the healthiest lifestyles was used as reference group.

## 4. Result

This cross-sectional study included 525 (212 men and 313 women) adults. Mean ± SD of age was $42.90 \pm 11.89$ years and for the BMI was $27.78 \pm 4.88$ kg/m$^2$. The majority of participants in the sample were female (60%) and 92.6% were married. The majority of participants had less than high School / diploma education level (70.9%). Additional information about participant's characteristics is presented in Table 3. According to our findings, 24.8% of people had a healthy dietary pattern and based on the physical activity guideline, 35.2% of the participants had low physical activity. Only 11.6% of the participants were smoker and 26 (5%) had a severe anxiety level.

### 4.1. Findings of the factor analysis

Using a factor analysis method, three dietary patterns were extracted and labeled according to the food groups with high loadings healthy pattern, unhealthy pattern, and traditional pattern which explained 22.66% contribution for whole variance (Table 2). Healthy dietary pattern was highly correlated with fruit, vegetables, fruit juice, dairies, and legumes intake. Unhealthy dietary pattern demonstrated high factor loadings for refined grains, fast foods, animal fats, pastries, soft drinks and chips. The traditional pattern was accompanied with high intake of fish, red meat, organ meat (heart, kidney, liver, and other animal organs), pickles, dry fruits, and nuts intake.

### 4.2. Findings of the latent-class analysis

Considering the four dichotomous indicators, life style patterns could be detected. LCA models with classes ranging from 1 to 6 were fitted and to determine the number of classes, the $G^2$ statistic test and its p-value, BIC, and AIC were calculated for each LCA model (Table 4). Based on the model's indices and interpretation of their results, we decided that a model with 3 latent classes was most appropriately fit. The results of the three LCA model displayed that differences between the observed and expected response pattern frequency were not statistically significant ($G^2 = 7.78$, df = 15, P-value = 0.941). Latent class prevalence and item-response probabilities for each indicator are presented in Table 5. The probability of membership in each latent class is expressed in the first section of Table 5. About 24% of participants were categorized in the class (I), 16.5% in the class (II), and 59.5% in class (III) lifestyles. These probabilities form the basis for interpreting and labeling the latent classes. The conditional probabilities of a "Yes" answer to each risk behavior for lifestyle are listed in Table 5. The possibility of a "No" answer can be calculated by subtracting the item response probabilities from 1. Latent 1$^{st}$ class included 126 individuals with a mean age of 45.25 years and characterized by a healthy dietary pattern with higher intake of vegetables, fruit, fruit juice, dairies, and legumes, moderate physical activity and low level of smoking. In contrast, those in latent 2$^{nd}$ class was characterized by a traditional dietary pattern with high intake of fish, red meat, organ meat, pickles, dry fruits, and nuts, low level of physical activity and as in 1$^{st}$ class low probability of smoking. Individuals in 3$^{rd}$ class similar to the members of the other two classes exhibited

Table 3. General demographic, anthropometric and biochemical parameters in participants.

| Variable | Total participants (N = 525) |
|---|---|
| Age (year) | 42.90[11.89] |
| Sex (male) | 212(40.38%) |
| Education | |
| Illiterate | 78(14.9%) |
| ≤ High school/diploma | 372(70.9%) |
| ≥ College degree | 75(14.3%) |
| Marital status | |
| Single | 39(7.4%) |
| Married | 486(92.6%) |
| Anxiety | |
| No anxiety | 262(49.9%) |
| Mild | 142(27%) |
| Moderate | 82(15.6%) |
| Severe | 26(5%) |
| Smoking | |
| Never | 464(88.4) |
| current and former smokers | 61(11.6%) |
| Weight | 73.91±13.18 |
| BMI (kg/m$^2$) | 27.78±4.88 |
| WC (cm) | 93.35±11.87 |
| WHR (cm) | 0.89±0.09 |
| DBP (mmHg) | 78.31±11.61 |
| SBP (mmHg) | 120.82±17.71 |
| TG (mg/dl) | 160.28±99.16 |
| TC (mg/dl) | 181.00±39.91 |
| HDL (mg/dl) | 43.87±10.21 |
| LDL (mg/dl) | 92.00±29.53 |
| FBS (mg/dl) | 89.00±23.96 |
| ALT (units/l) | 18.54±9.49 |
| AST (units/l) | 21.03±6.13 |
| HB (g/dl) | 14.01±1.75 |
| Ferritin (μg/L) | 90.46±162.72 |
| Serum vit D (ng/ml) | 24.15±20.41 |

BMI, body mass index; WC, waist circumference; WHR, Waist-to-hip ratio; DPB, diastolic blood pressure; SBP, systolic blood pressure; TG, triglyceride; TC, total cholesterol; LDL, low density lipoprotein cholesterol; HDL-C: high density lipoprotein cholesterol; FBS: fasting blood glucose; ALT, alanine amino transferase; AST, aspartate amino transferase; HB, hemoglobin. Atherogenic index = (Total cholesterol–HDL-C) / HDL-C. Discrete and continuous variables data are presented as number (percent) and mean ± SD.

lower probabilities of smoking and were likely to report that they had engaged in low level physical activity and an unhealthy dietary pattern characterized by higher intake of refined grains, fast foods, animal fats, pastries, soft drinks and chips. Table 6 presents the biochemical and demographic characteristics of participants, according to the latent class membership. There were significant differences in the education levels between the classes and compared with the 1$^{st}$ class and 3$^{rd}$ class, a higher proportion of participants in 2$^{nd}$ class had an illiterate education level (P = 0.03) and no significant difference in marital status between the three

**Table 4. Comparison of LCA models with different latent classes based on model selection statistics.**

| No. LCA | No. parametersestimated | G² | df | P-value | Log likelihood | AIC | BIC |
|---|---|---|---|---|---|---|---|
| 1 | 6 | 38.59 | 29 | 0.120 | -1412.87 | 50.52 | 76.18 |
| 2 | 13 | 19.37 | 22 | 0.636 | -1403.32 | 45.44 | 100.92 |
| 3 | 20 | 7.78 | 15 | 0.941 | -1397.43 | 47.83 | 133.08 |
| 4 | 27 | 5.68 | 8 | 0.675 | -1398.44 | 59.68 | 174.87 |
| 5 | 34 | 34.56 | 1 | 0.068 | -1397.28 | 71.39 | 216.51 |

Df: degrees of freedom, BIC: Bayesian information criterion, AIC: Akaike information criterion, LCA = latent class analysis

classes was obtained (P = 0.55). There were significant differences in BMI, FBS, HB and WHR across different classes and the participants in the 3rd class had significantly higher BMI and FBS compared to other classes (P = 0.02). Participants in the 1st class had significantly higher HB compared to other classes (P = 0.02) and individuals in the 2nd class had significantly higher WHR compared to other classes (P = 0.02).The three study classes did not differ significantly in mean age, weight, WC, SBP, DBP, TG, TC, LDL, HDL, ferritin, ALT, AST, and serum vitamin D (Table 6).

## 5. Discussion

To the best of our knowledge, the current community-based study was the first report to demonstrate the lifestyle patterns using LCA in the northwest of Iran. According to the findings, participants classified in three sub-classes according to their physical activity level, dietary patterns and smoking. Also, three dietary patterns (healthy, unhealthy and traditional) were emerged using factor analysis. Individuals in 1st class had the following characteristics: 47.4% of them had moderate physical activity, were not cigarette smokers (77%), and had healthy dietary pattern (58.9%). While most of the individuals in 2nd class had low physical activity level (82%), smoking (20.3%) with the traditional diet as the dominant dietary pattern (83.2%). The largest class (3rd class) reflected 59.5% of the sample and individuals had low physical activity (44.8%) with 1.6% prevalence of smoking with the unhealthy dietary pattern as preferred dietary pattern (51.5%). Physical activity levels and dietary pattern were the best indicators for the differentiation of classes. While the smoking variable indicated similar probabilities in three classes. Irrespective of latent class membership, the probability of cigarette smoking was

**Table 5. The three latent classes models of lifestyle.**

| Variable | 1st class | 2nd class | 3rd class |
|---|---|---|---|
| Prevalence (%) | 0.240 | 0.165 | 0.595 |
| Items | | | |
| Smoking | | | |
| No | **0.770** | **0.797** | **0.984** |
| Yes | 0.230 | 0.203 | 0.016 |
| Physical inactivity | | | |
| High | 0.321 | 0.106 | 0.265 |
| Moderate | **0.474** | 0.074 | 0.287 |
| Low | 0.205 | **0.820** | **0.448** |
| Nutrition status | | | |
| Healthy diet | **0.589** | 0.000 | 0.340 |
| Unhealthy diet | 0.137 | 0.168 | **0.515** |
| Traditional diet | 0.274 | **0.832** | 0.145 |

**Table 6. Anthropometric, biochemical and demographic parameters according to three latent classes.**

| Variable | 1st class | 2nd class | 3rd class | P-value |
|---|---|---|---|---|
| Age (year) | 45.25±12.07 | 42.50±11.80 | 42.55±11.94 | 0.21 |
| Weight (kg) | 75.56±13.65 | 70.92±11.98 | 74.16±13.30 | 0.09 |
| BMI (kg/m$^2$) | 27.87±5.24 | 26.26±3.55 | 28.07±4.96 | 0.02 |
| WC (cm) | 96.00±12.67 | 94.08±10.85 | 93.92±11.85 | 0.40 |
| WHR (m) | 0.87±0.07 | 0.91±0.17 | 0.89±0.8 | 0.02 |
| SBP (mmHg) | 123.96 ±17.74 | 121.33±21.46 | 122.09±16.82 | 0.68 |
| DBP (mmHg) | 78.34±10.34 | 79.87±14.33 | 79.48±11.02 | 0.47 |
| TG (mg/dl) | 157.49±96.62 | 145.89±75.87 | 146.89±75.87 | 0.67 |
| TC (mg/dl) | 183.02±36.18 | 186.84±40.66 | 181.24±40.36 | 0.56 |
| LDL-C (mg/dl) | 93.84±26.76 | 95.16±32.81 | 92.61±29.59 | 0.79 |
| HDL-C (mg/dl) | 43.85±10.81 | 44.10±9.98 | 44.29±10.16 | 0.43 |
| FBS (mg/dl) | 86.94±14.20 | 83.86±25.31 | 91.35±23.70 | 0.02 |
| Ferritin(µg/L) | 83.69±59.61 | 91.66±110.79 | 92.48±186.66 | 0.88 |
| HB(g/dl) | 14.57±1.90 | 13.59±2.06 | 13.94±1.68 | 0.02 |
| ALT (units/l) | 18.52±8.46 | 16.76±7.94 | 19.12±10.00 | 0.16 |
| AST (units/l) | 20.93±5.30 | 22.25±5.25 | 20.96±6.48 | 0.26 |
| Serum Vit D (ng/ml) | 24.84±18.70 | 22.19±23.83 | 24.58±20.46 | 0.63 |
| Education | | | | 0.03 |
| Illiterate | 9(12.7) | 14(20.3) | 53(14.5) | |
| ≤ High school/diploma | 54(76) | 52(75.4) | 250 (68.3) | |
| ≥ College degree | 8(11.3) | 3(4.3) | 63(17.2) | |
| Marital status | | | | 0.55 |
| Married | 66(91.5) | 65(95.7) | 337(92.1) | |
| Anxiety | | | | 0.01 > |
| No anxiety | 19(27.1%) | 36(53.7%) | 207(58%) | |
| Mild | 7(10%) | 20(29.9) | 115(32.2%) | |
| Moderate | 31(44.3%) | 9(13.4%) | 31(8.7%) | |
| Severe | 13(18.6%) | 2(3%) | 4(1.1%) | |

BMI, body mass index; WC, waist circumference; WHR, Waist-to-hip ratio; DPB, diastolic blood pressure; SBP, systolic blood pressure; TG, triglyceride; TC, total cholesterol; LDL, low density lipoprotein cholesterol; HDL-C: high density lipoprotein cholesterol; FBS: fasting blood glucose; ALT, alanine amino transferase; AST, aspartate amino transferase; HB, hemoglobin. Atherogenic index = (Total cholesterol–HDL-C) / HDL-C. P-value for categorical variables based on Chi-Square test and for continuous variables based on ANOVA. Discrete and continuous variables data are presented as number (percent) and mean ± SD.

relatively low across the total sample. Therefore, cigarette smoking might not be good indicators for predicting unhealthy or healthy lifestyles in our study. Previous research showed that considering the co-occurrence of unhealthy lifestyle behaviors together such as the combination of smoking and low physical activity [48], and the combination of smoking and low vegetable and fruit consumptions [49], are beneficial approaches for prevention of various diseases [50, 51]. In recent years, several studies have evaluated the patterns of lifestyle behaviors among adults. However, these studies are using different analytical methods, different lifestyle behaviors and are conducted in different populations make the comparison of results difficult. In 2018, Akbarpour S et al [52] evaluated the lifestyle behavior patterns among Iranian population using self-organizing map. They identified seven clusters of lifestyle patterns. Class 1 as the healthiest lifestyle, class 2 with the high intake of fast food, salt, and soft drinks, class 3 with no leisure physical activity, class 4 with smoking, consumption of alcohol and soft drinks, class 5 with absence of physical activity and less oil and salt consumption, class 6 with no

consumption of dairy products, and class 7 with unhealthy lifestyles; they found that, persons who were in unhealthy lifestyle classes were commonly less educated and more self-employed. In another study, Laska et al using LCA identified four latent classes for men and four latent classes for women [53]. They linked a collection of behaviors such as physical activity, diet, stress, tobacco use, alcohol consumption, risky sexual behaviors, and sleep. In a cross-sectional study, authors found that approximately 50% of study members had at least 3 unhealthy lifestyle behaviors and nearly 80% of them had multiple unhealthy behaviors [54]. In study by Chiolero A [55], smokers had high alcohol intake, low fruits/vegetable consumption and low physical activity level than non-and ex-smokers. These results further confirm the inter-relation between different risk behaviors, as previously reported by previous studies [56, 57]. In the current study, 35.2% of participants had low physical activity while 30.9% and 33.9% of participants had moderate and high physical activity, respectively. One other important result of the current study was that only 24% of participants were categorized in 1st class and those in this class had higher intake of vegetables, fruit, fruit juice, dairies, and legumes. 1st class was categorized as and almost low-risk lifestyle class with healthy dietary pattern and moderate physical activity. Charreire H et al [58] found significant positive associations between healthy dietary pattern and leisure time physical activity in women and men. Similarly, in the study by Gillman MW et al [59] increased physical activity was significantly related to higher intake of healthy and fresh foods. In other study, high levels of physical activity were related with higher intake of vegetables, dairy products, fruits, and milk among both girls and boys [60]. The intake of dairies, fruits and vegetables and physical activity was low in class 2. So, one of the most preventive interventions in this group focuses on enhancing physical activity and promoting healthy diet. The 3rd class includes large segments of adults (59.5%) and the probability of smoking and following a healthy dietary pattern is 1.6% and 34%, respectively but in this class, probability of following the unhealthy dietary pattern is more than the healthy dietary pattern. In the study by Suh S-Y et al [61] smoking in Korean men was inversely related with a healthy dietary pattern as showed in the 'fruits, dairy products, and potatoes' and 'sea food and vegetables' patterns. Additionally, in the present study, our findings revealed the significant differences between subgroups in terms of BMI, WHR, HB and FBS. Participants in the 3rd class with high adherence to unhealthy dietary pattern and low level of physical activity had significantly higher BMI and FBS compared to other classes. This may be due to the high intake of sweetened drinks and refined grains in unhealthy dietary pattern [62, 63]. Walsh EI et al [64] found no significant relationship between adherence to unhealthy dietary pattern and fasting blood glucose; while in another study adherence to unhealthy dietary pattern was related to higher BMI and hyperlipidemia [65]. In the study by Gutiérrez-Pliego LE et al [66] individuals in the highest tertiles of unhealthy dietary pattern revealed higher BMI compared with individuals in the highest tertile of the prudent dietary pattern. Similar findings were also reported in other studies [67–69]. The current study has several limitations. First, the design of this study was cross-sectional and causality could not be addressed. Second, the information of dietary intake, smoking and anxiety status were collected as self-reported data and might be a source of bias although all of the questionnaires validity and reliability were confirmed previously. Although the present study considered several factors in relation to high-risk behaviors among adults, there may be other variables that were not addressed. Although a relatively large sample size of the current study and including both females and males in almost all age groups are potent strengths of the current research.

## 6. Conclusion

In conclusions, our findings identified three classes of life style patterns in a community-based study in Northwest of Iran. Also, three main dietary patterns including healthy, traditional,

and unhealthy dietary patterns were extracted using factor analysis. Using LCA in the current report revealed the merits of recognizing the heterogeneity that exists in participants. Three classes derived from the person-centered method characterized by different behavior. Health-care professionals should be aware of associations between different lifestyle risk factors and health promotion strategies should further focus on multiple behaviors at the same time.

## Supporting information

**S1 Dataset.**
(SAV)

**S1 File.**
(PDF)

**S2 File.**
(PDF)

**S3 File.**
(PDF)

**S4 File.**
(PDF)

**S5 File.**
(PDF)

**S6 File.**
(PDF)

**S7 File.**
(PDF)

**S8 File.**
(PDF)

## Acknowledgments

We thank all of the study participants.

## Author Contributions

**Conceptualization:** Mahdi Vajdi, Asghar Mohammad Pour Asl, Mahdieh Abbasalizad Farhangi.

**Data curation:** Asghar Mohammad Pour Asl, Mahdieh Abbasalizad Farhangi.

**Formal analysis:** Asghar Mohammad Pour Asl, Mahdieh Abbasalizad Farhangi.

**Funding acquisition:** Mahdieh Abbasalizad Farhangi.

**Investigation:** Mahdi Vajdi, Leila Nikniaz, Asghar Mohammad Pour Asl, Mahdieh Abbasali-zad Farhangi.

**Methodology:** Mahdi Vajdi, Leila Nikniaz, Mahdieh Abbasalizad Farhangi.

**Software:** Asghar Mohammad Pour Asl.

**Supervision:** Leila Nikniaz, Mahdieh Abbasalizad Farhangi.

**Validation:** Leila Nikniaz, Asghar Mohammad Pour Asl, Mahdieh Abbasalizad Farhangi.

**Visualization:** Mahdi Vajdi, Asghar Mohammad Pour Asl, Mahdieh Abbasalizad Farhangi.

**Writing – original draft:** Mahdi Vajdi, Mahdieh Abbasalizad Farhangi.

**Writing – review & editing:** Mahdi Vajdi, Leila Nikniaz, Asghar Mohammad Pour Asl, Mahdieh Abbasalizad Farhangi.

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
