## [Decision Letter · Decision Letter 0]

13 Feb 2020

PONE-D-19-32691

Lifestyle patterns and their nutritional, socio-demographic and psychological determinants in a community-based study: A mixed approach of latent class and factor analyses

PLOS ONE

Dear Mahdieh Abbasalizad Farhangi

Thank you for submitting your manuscript to PLOS ONE. After careful consideration, we feel that it has merit but does not fully meet PLOS ONE’s publication criteria as it currently stands. Therefore, we invite you to submit a revised version of the manuscript that addresses the points raised during the review process.

We would appreciate receiving your revised manuscript by Mar 29 2020 11:59PM. To enhance the reproducibility of your results, we recommend that if applicable you deposit your laboratory protocols in protocols.io, where a protocol can be assigned its own identifier (DOI) such that it can be cited independently in the future. For instructions see: http://journals.plos.org/plosone/s/submission-guidelines#loc-laboratory-protocols

We look forward to receiving your revised manuscript.

Kind regards,

Paul Gerard Shiels, BA (Mod)., PhD

Academic Editor

PLOS ONE

Journal Requirements:

4. Thank you for stating the following on the title page of your manuscript:

"The work has been granted by Research Undersecretary of Tabriz University of Medical Sciences (Registration number: IR.TBZMED.REC.1398.077)."

5 Please include your tables as part of your main manuscript and remove the individual files. Please note that supplementary tables (should remain/ be uploaded) as separate "supporting information" files

Reviewers' comments:

Reviewer's Responses to Questions

**Comments to the Author**

1. Is the manuscript technically sound, and do the data support the conclusions?

Reviewer #1: Partly

Reviewer #2: No

2. Has the statistical analysis been performed appropriately and rigorously? 

Reviewer #1: Yes

Reviewer #2: No

3. Have the authors made all data underlying the findings in their manuscript fully available?

Reviewer #1: Yes

Reviewer #2: No

4. Is the manuscript presented in an intelligible fashion and written in standard English?

Reviewer #1: Yes

Reviewer #2: No

5. Review Comments to the Author

Reviewer #1: This paper is well written and uses a novel data set based on 525 adults from East Azarbaijan, Iran. The main issue I have with the paper is that it is highly unusual to utilise a mental health variable such as anxiety as a 'lifestyle behaviour'. Lifestyles behaviours are normally smoking, dietary intake, physical activity and alcohol. Anxiety is often associated with lifestyle behaviours such as smoking, diet and physical activity but is not normally included as a 'lifestyle behaviour' itself.

I think the authors need to revise their research questions and re-analyse their data to exclude anxiety (although they may wish to explore the association between anxiety and smoking, diet and physical activity).

Reviewer #2: 

The authors state that the data underlying the study is fully available without restriction, however the following question looking for details of where it is available has not been completed correctly.

The source of funding statement needs re-phrased.

There is a massive contradiction in the introduction, the authors state that other studies focus on single lifestyle factors in isolation then continue to cite several articles that look at these in combination or show correlations between them.

A copy of the questions/questionnaire should be presented with the manuscript so that the validity of the questions can be assessed by the reader.

The authors state that individuals with physical disability, severe illness and a few other conditions were excluded but have not provided figures for these exclusions, where they excluded from the larger study and thus were not present in the cohort or where they excluded precisely from this study?

The restriction to Iranian nationals limits the results and conclusions to this narrow focus - invalidating most of the global seeming claims in the abstract, this is hinted at in the conclusions but should be clearly and concisely stated.

It is stated that 2 blood pressure measurements were taken and then averaged, the standard deviations should also be stated to ensure these were not skewed by other factors e.g. stress.

No information about collection of blood samples has been provided, i.e. was there anti-coagulant added to the blood or was the sample clotted naturally? Were any other additives added to the blood? Catalogue numbers should be included.

Alanine amino transferase is ALT, not AST and no definition is provided for AST. The ultraviolet measurement method is not described or referenced.

"Binary indicators (healthy versus unhealthy behaviour) were created based on existing recommendations and/or propositions by the literature." Needs to be properly referenced with examples.

Anxiety status, defined in methods as mild, moderate and severe; but referred to elsewhere as low, moderate and high - consistency in description.

Under smokers there is no provision for passive smoking, former smokers or heavy/light smokers.

Descriptive statistics - MANOVA should be used for multiple testing not one-way ANOVA and there is no correction for multiple testing.

Factor analysis - 21 food groups are mentioned, then 23? need to be accurate and consistent. Part of this section repeats itself.

Latent Class Analysis - anxiety status is listed as a dichotomous factor, but under description there are 3 categories defined - clarification needed as to what this represents. The article relies very heavily on latent class analysis (LCA), however a detailed description of the process and references are required to support its use and implementation. Adjustment for multiple comparisons is needed, if any were performed these have not been described. "related to the signs" needs to be clearly defined. This whole section is lacking in references and description.

The use of "Western dietary pattern" is inaccurate and not an accepted description while this may be an accepted local colloquialism perhaps it would be more accurate, descriptive and appropriate to refer to "unhealthy diet pattern".

The three groupings or classes are severely unbalanced casting doubt on the statistical significance of these and on the study design.

All 3 groups had low probabilities of smoking and anxiety disorder - if these were particularly low then they should have been excluded as factors in the analysis, or the analysis should focus on proportionate levels between the groups and the whole cohort. Given the "low" status of these factors in all groups it can't be that these were used to define the categories or that they had any impact at all.

Traditional dietary pattern: how organ and red meat was defined as this can be categorised as unprocessed and process red meat e.g., pate will be both red meat and organ meat, but it is processed. No list of processed meat was included in table 1. It is generally accepted that processed vs unprocessed meats are a risk/discriminant factor when examining diet, this has not been mentioned let alone addressed in this manuscript

Definition of the 150 clusters is required.

More detailed definition of metabolic equivalents (MET -minutes/week) is required, Anxiety status needs expansion and more detailed description.

It would be nice to include some visual representation of data normality e.g., the frequency distribution (histogram), stem-and-leaf plot, boxplot, P-P plot (probability-probability plot), or Q-Q plot (quantile-quantile plot). Similarly a visual representation of the overall analysis would be beneficial as would the actual PCA plots from the PCA analysis.

No description of meaning of factor analysis with varimax rotation and why PCA, more basic exploratory analysis was used?

Both methods allow reducing the dimensionality of datasets; however EEA is designed with the objective to identify certain unobservable factors from the observed variables, whereas PCA at best, PCA provides an approximation to the required factors.

How was the factor score calculated.

"Most of them had moderate physical activity (45%)" this is less than half, it is inaccurate to state "most"

The article in general massively over interprets the results and makes large claims that are unsupported by the data, the large spread of factors across all groups indicates that these are unlikely to be significant factors in their own right within this cohort and therefore only very specific combinations may be valid but these were not explored at all.

The supposed healthy group actually had the highest levels of smoking and anxiety, comparisons between groups and to the whole cohort would be very telling in this instance.

Was gender investigated as a discriminatory factor?

The conclusions are far fetched and unsupported, especially given the severe limitations of the study.

6. PLOS authors have the option to publish the peer review history of their article (what does this mean?). If published, this will include your full peer review and any attached files.

Reviewer #1: No

Reviewer #2: No

---

## [Author Response · Author response to Decision Letter 0]

28 Mar 2020

Ms. Ref. No.: PONE-D-19-32691 20 March, 2020 

Title: Lifestyle patterns and their nutritional, socio-demographic and psychological determinants in a community-based study: A mixed approach of latent class and factor analyses

Dear Professor Paul Gerard Shiels

I hope you are well. Thank you very much for your close attention to our submitted manuscript. I have revised the manuscript according to the reviewers’ comments. Also, following is our response to the reviewer’s comments. The comments by reviewers are presented in black color and the responses from the authors are presented in blue color. The comments which were given alongside the papers of the manuscript have been corrected and marked with yellow color. 

Yours sincerely,

Mahdieh Abbasalizad Farhangi, Ph.D. 

Associate Professor 

Drug Applied Research Center, 

Tabriz University of Medical Sciences, 

Tabriz, Iran

Journal Requirements:

1. Please ensure that your manuscript meets PLOSONE's style requirements, including those for file naming. The PLOS ONE style templates can be found at http://www.journals.plos.org/plosone/s/file?id=wjVg/PLOSOne_formatting_sample_main_body.pdf and http://www.journals.plos.org/plosone/s/file?id=ba62/POSOne_formatting_sample_title_authors_affiliations.pdf

Action or explanation: Thank you for your comment. The correction has been made as requested.

Action or explanation: Thank you for your comment. The correction has been made as requested. 

Action or explanation: Thank you for your comment. The correction has been made as requested.

Action or explanation: Thank you for your comment. The correction has been made as requested.

4. Thank you for stating the following on the title page of your manuscript:

"The work has been granted by Research Undersecretary of Tabriz University of Medical Sciences (Registration number: IR.TBZMED.REC.1398.077)."

Action or explanation: Thank you for your comment. The correction has been made as requested.

5 Please include your tables as part of your main manuscript and remove the individual files. Please note that supplementary tables (should remain/ be uploaded) as separate "supporting information" files

Action or explanation: Thank you for your comment. The correction has been made as requested.

Important: If there are ethical or legal restrictions to sharing your data publicly, please explain these restrictions in detail. Please see our guidelines for more information on what we consider unacceptable restrictions to publicly sharing data: http://journals.plos.org/plosone/s/data-availability#loc-unacceptable-data-access-restrictions. Note that it is not acceptable for the authors to be the sole namedindividuals responsible for ensuring data access. We will update your Data Availability statement to reflect the information you provide in your cover letter.

Action or explanation: Thank you for your comment. The correction has been made as requested.

Reviewers' comments:

Reviewer's Responses to Question

Comments to the Author

1. Is the manuscript technically sound, and do the data support the conclusions?

Reviewer#1: Partly

Reviewer #2: No

2. Has the statistical analysis been performed appropriately and rigorously?

Reviewer#1: Yes

Reviewer #2: No

3. Have the authors made all data underlying the findings in their manuscript fully available?

Reviewer#1: Yes

Reviewer #2: No 

4. Is the manuscript presented in an intelligible fashion and written in standard English?

Reviewer #1: Yes

Reviewer #2: No

5. Review Comments to the Author

Reviewer #1: 

This paper is well written and uses a novel data set based on 525 adults from East Azarbaijan, Iran. The main issue I have with the paper is that it is highly unusual to utilize a mental health variable such as anxiety as 'lifestyle behavior'. Lifestyles behaviors are normally smoking, dietary intake, physical activity and alcohol. Anxiety is often associated with lifestyle behaviors such as smoking, diet and physical activity but is not normally included as a 'lifestyle behavior' itself. I think the authors need to revise their research questions and re-analyse their data to exclude anxiety (although they may wish to explore the association between anxiety and smoking, diet and physical activity).

Action or explanation: Thank you for your comment. You are absolutely right. Anxiety status was excluded from the latent class analysis. Please see the revised manuscript.

Reviewer #2:

The authors state that the data underlying the study is fully available without restriction; however the following question looking for details of where it is available has not been completed correctly.

Action or explanation: Thank you for your comment. The correction has been made as requested.

The source of funding statement needs re-phrased.

Action or explanation: Thank you for your comment. The correction has been made as requested.

There is a massive contradiction in the introduction, the author's state that other studies focus on single lifestyle factors in isolation then continue to cite several articles that look at these in combination or show correlations between them.

Action or explanation: Thank you for your comment. We corrected it in the text. Please see the revised manuscript.

A copy of the questions/questionnaire should be presented with the manuscript so that the validity of the questions can be assessed by the reader.

Action or explanation: Thank you for your comment. The correction has been made as requested.

The authors state that individuals with physical disability, severe illness and a few other conditions were excluded but have not provided figures for these exclusions, where they excluded from the larger study and thus were not present in the cohort or where they excluded precisely from this study?

Action or explanation: Thank you for your comment. Subjects with physical disability, severe chronic illness requiring bed rest, mental disability, active liver injury, and pregnant women were not included in our study.

The restriction to Iranian nationals limits the results and conclusions to this narrow focus - invalidating most of the global seeming claims in the abstract, this is hinted at in the conclusions but should be clearly and concisely stated.

Action or explanation: Thank you for your comment. The paper abstract was corrected. Please see the revised manuscript.

It is stated that 2 blood pressure measurements were taken and then averaged, the standard deviations should also be stated to ensure these were not skewed by other factors e.g. stress.

Action or explanation: Thank you for your comment. The standard deviation of blood pressure is listed in Table 3.

No information about collection of blood samples has been provided i.e. was there anti-coagulant added to the blood or was the sample clotted naturally? Were any other additives added to the blood? Catalogue numbers should be included.

Action or explanation: Thank you for your comment. The correction has been made as requested. Please see the revised manuscript .After a 12–14 h overnight fasting, 10 ml venous blood samples were collected from all participants and placed in vacutainer/ siliconized test tubes without anticoagulant for biochemical determinations.

Alanine amino transferase is ALT, not AST and no definition is provided for AST. The ultraviolet measurement method is not described or referenced.

Action or explanation: The corrections had been made as requested. Please see the revised manuscript.

"Binary indicators (healthy versus unhealthy behavior) were created based on existing recommendations and/or propositions by the literature." Needs to be properly referenced with examples.

Action or explanation: Thank you for your comment. The corrections had been made as requested. Please see the revised manuscript.

Anxiety status, defined in methods as mild, moderate and severe; but referred to elsewhere as low, moderate and high - consistency in description.

Action or explanation: Thank you for your comment. The manuscript had been revised accordingly. Please see the revised manuscript. Scores of 0-5, 6-9, 10-14 and ≥15 specify normal, low, moderate and high symptoms of anxiety, respectively.

Under smokers there is no provision for passive smoking, former smokers or heavy/light smokers.

Action or explanation: Thank you for your comment. Tobacco consumption was defined as daily smoking of at least one cigarette, cigar, pipe or cigarillo within the last 30 days and a binary indicator for smoking was created (never smoked versus current and former smokers). Never smoked and current and former smokers are listed in Table 3.

Descriptive statistics - MANOVA should be used for multiple testing not one-way ANOVA and there is no correction for multiple testing.

Action or explanation: 

Factor analysis- 21 food groups are mentioned, then 23? Need to be accurate and consistent. Part of this section repeats itself.

Action or explanation: Thank you for your comment. We sincerely apologize for the mistake. The manuscript had been revised accordingly. Please see the revised manuscript.

Latent Class Analysis - anxiety status is listed as a dichotomous factor, but under description there are 3 categories defined - clarification needed as to what this represents. The article relies very heavily on latent class analysis (LCA), however a detailed description of the process and references are required to support its use and implementation. Adjustment for multiple comparisons is needed, if any were performed these have not been described. "related to the signs" needs to be clearly defined. This whole section is lacking in references and description.

Action or explanation: The manuscript had been revised accordingly. Please see the revised manuscript. Thank you for your comment. Anxiety status was excluded from the latent class analysis. Scores of 0-5, 6-9, 10-14 and ≥15 specify normal, low, moderate and high symptoms of anxiety, respectively. The LCA is used to identify the latent classes and unknown patterns based on a set of observed variables from multivariate classified data.

Theuse of "Western dietary pattern" is inaccurate and not an accepted description while this may be an accepted local colloquialism perhaps it would be more accurate, descriptive and appropriate to refer to "unhealthy diet pattern".

Action or explanation: Thank you for your comment. The manuscript had been revised accordingly. Please see the revised manuscript

The three groupings or classes are severely unbalanced casting doubt on the statistical significance of these and on the study design.

Action or explanation: Thank you for your comment

All 3 groups had low probabilities of smoking and anxiety disorder - if these were particularly low then they should have been excluded as factors in the analysis, or the analysis should focus on proportionate levels between the groups and the whole cohort. Given the "low" status of these factors in all groups it can't be that these were used to define the categories or that they had any impact at all.

Action or explanation: Thank you for your comment. You are right. Anxiety status was excluded from the latent class analysis.

Traditional dietary pattern: how organ and red meat was defined as this can be categorized as unprocessed and process red meat e.g., pate will be both red meat and organ meat, but it is processed. No list of processed meat was included in table 1. It is generally accepted that processed vs unprocessed meats are a risk/discriminant factor when examining diet, this has not been mentioned let alone addressed in this manuscript.

Action or explanation: Thank you for your comment. For food pattern determination, a quantitative food frequency questionnaire (FFQ) was used. The FFQ included 80 questions which was developed and validated previously and reference portions were defined based on the most reported portion sizes in the 24-h recall. Based on the validation of the questionnaire, the people in this region use very little processed meat. Therefore processed meats are not included in our FFQ. 

Ref.

Nikniaz L, Tabrizi J, Sadeghi-Bazargani H, Farahbakhsh M, Tahmasebi S, Noroozi S. Reliability and relative validity of short-food frequency questionnaire. British Food Journal. 2017.

Definition of the 150 clusters is required.

Action or explanation: Thank you for your comment. The manuscript had been revised accordingly. Please see the revised manuscript. This study conducted by probability proportional to size (PPS) multistage stratified cluster sampling. Clusters comprise one to several blocks or parts of blocks. Blocks were usually attached buildings.

More detailed definition of metabolic equivalents (MET -minutes/week) is required, Anxiety status needs expansion and more detailed description.

Action or explanation: Thank you for your comment. The manuscript had been revised accordingly. Please see the revised manuscript.

It would be nice to include some visual representation of data normality e.g., the frequency distribution (histogram), stem-and-leaf plot, boxplot, P-P plot (probability-probability plot), or Q-Q plot (quantile-quantile plot). Similarly a visual representation of the overall analysis would be beneficial as would the actual PCA plots from the PCA analysis.

Action or explanation: Thank you for your comment.

No description of meaning of factor analysis with varimax rotation and why PCA, more basic exploratory analysis was used?

Action or explanation: Thank you for your comment. The manuscript had been revised accordingly. Please see the revised manuscript. Distinguishing between overlapping components is a main reason why PCA has been used. The varimax rotation was used because it has the potential to minimize the number of components. The PCA is a method for finding a lower-dimensional representation, which accounts for the majority of the variance in the features. Factor analysis with varimax rotation is performed to develop a more interpretable solution. The rotated solution uses a PCA of reduced dimensionality as the starting point. In general, rotation needed because it makes the factor structures meaningful and easily interpretable.

Both methods allow reducing the dimensionality of datasets; however EEA is designed with the objective to identify certain unobservable factors from the observed variables, whereas PCA at best, PCA provides an approximation to the required factors.

How was the factor score calculated?

Action or explanation: Thank you for your comment. Please see the revised manuscript. The factor score for each pattern was calculated by summing consumption of all food groups weighted by their factor loadings.

"Most of them had moderate physical activity (45%)" this is less than half, it is inaccurate to state "most"

Action or explanation: Thank you for your comment. We sincerely apologize for the mistake. Please see the revised manuscript.

The article in general massively over interprets the results and makes large claims that are unsupported by the data, the large spread of factors across all groups indicates that these are unlikely to be significant factors in their own right within this cohort and therefore only very specific combinations may be valid but these were not explored at all.

Action or explanation: Thank you for your comment. 

The supposed healthy group actually had the highest levels of smoking and anxiety, comparisons between groups and to the whole cohort would be very telling in this instance.

Action or explanation: Thanks for the comment. You are absolutely right. In our study, participants in the first class have a healthy dietary pattern and moderate physical activity, but we can't say that they have a perfectly healthy lifestyle.

Was gender investigated as a discriminatory factor?

Action or explanation: Thank you for your comment 

The conclusions are far fetched and unsupported, especially given the severe limitations of the study.

 Action or explanation: Thank you for your comment. The manuscript had been revised accordingly. Please see the revised manuscript

6. PLOS authors have the option to publish the peer review history of their article (what does this mean?). If published, this will include your full peer review and any attached files.

Do you want your identity to be public for this peer review? For information about this choice, including consent withdrawal, please see our Privacy Policy.

Reviewer#1: No

Reviewer#2: No

---

## [Decision Letter · Decision Letter 1]

6 Jul 2020

Lifestyle patterns and their nutritional, socio-demographic and psychological determinants in a community-based study: A mixed approach of latent class and factor analyses

PONE-D-19-32691R1

Dear Dr.  Farhangi 

We’re pleased to inform you that your manuscript has been judged scientifically suitable for publication and will be formally accepted for publication once it meets all outstanding technical requirements.

Kind regards,

Paul Gerard Shiels, BA (Mod)., PhD

Academic Editor

PLOS ONE

Additional Editor Comments (optional):

Reviewers' comments:

Reviewer's Responses to Questions

**Comments to the Author**

1. If the authors have adequately addressed your comments raised in a previous round of review and you feel that this manuscript is now acceptable for publication, you may indicate that here to bypass the “Comments to the Author” section, enter your conflict of interest statement in the “Confidential to Editor” section, and submit your "Accept" recommendation.

Reviewer #1: All comments have been addressed

2. Is the manuscript technically sound, and do the data support the conclusions?

Reviewer #1: Yes

3. Has the statistical analysis been performed appropriately and rigorously? 

Reviewer #1: Yes

4. Have the authors made all data underlying the findings in their manuscript fully available?

Reviewer #1: Yes

5. Is the manuscript presented in an intelligible fashion and written in standard English?

Reviewer #1: Yes

6. Review Comments to the Author

Reviewer #1: (No Response)

7. PLOS authors have the option to publish the peer review history of their article (what does this mean?). If published, this will include your full peer review and any attached files.

Reviewer #1: No

---

## [Editor Report · Acceptance letter]

8 Jul 2020

PONE-D-19-32691R1 

Lifestyle patterns and their nutritional, socio-demographic and psychological determinants in a community-based study: A mixed approach of latent class and factor analyses 

Dear Dr. Abbasalizad Farhangi:

I'm pleased to inform you that your manuscript has been deemed suitable for publication in PLOS ONE. Congratulations! Your manuscript is now with our production department. 

Kind regards, 

on behalf of

Professor Paul Gerard Shiels 

Academic Editor

PLOS ONE